# Place susceptibility index mapping at local government scale from population-based survey for Sub-Saharan Africa

Olanrewaju Lawal[1]*, Mmedorenyin Okon[1], Natalia Blanco[2,3], Christina Riley[4], James Onyemata[5], Anna Winters[4], Timothy O'Connor[6], Chenfeng Xiong[7], Alash'le Abimiku[2,5,8], Manhattan Charurat[2,8,9], Kristen A. Stafford[2,8], for The INFORM Africa Study Group¶

1 Department of Geography and Environmental Management, Faculty of Social Sciences University of Port Harcourt, Port Harcourt, Nigeria, 2 University of Maryland School of Medicine, Baltimore, Maryland, United States of America, 3 Institute of Human Virology, University of Maryland Baltimore, School of Medicine, Baltimore, Maryland, United States of America, 4 Akros, Lusaka, Zambia, 5 Institute of Human Virology Nigeria, International Research Center of Excellence, Abuja, Nigeria, 6 Institute of Genomic Sciences, University of Maryland Baltimore, Baltimore, Maryland, United States of America, 7 Department of Civil and Environmental Engineering, College of Engineering, Villanova University, Villanova, Pennsylvania, United States of America, 8 Division of Global Health Sciences, Department of Epidemiology and Public Health, University of Maryland School of Medicine, Baltimore, Maryland, United States of America, 9 Division of Geographic Medicine, Department of Medicine. University of Maryland School of Medicine, Baltimore, Maryland, United States of America

¶ Complete membership of the group can be found in the Acknowledgments
* olanrewaju.lawal@uniport.edu.ng

## Abstract

The Place Susceptibility Index (PSI) has the potential to be a critical tool in effectively managing infectious disease outbreaks or natural disasters in Africa. However, PSI availability for the African continent is limited and often, when available, is only at the national or regional level. Thus, lacking the required details to support locally relevant decision-making to support such activities. Here, outlined the method for modeling PSI at the 3rd-order administrative level for selected African Countries. This method combined Bayesian spatial statistical modeling with the utilization of the Population-based HIV Impact Assessments (PHIA) data and geospatial covariates. Using the Jenks Classification, substantial variations in PSI classes across the countries were observed. Across the 10 countries, about 45% of the spatial units were categorized as the low and relatively low susceptibility classes, while around 31% belonged to the high and very high classes. Botswana had 17% of the spatial units classified as high or very high, while Zambia had as many as 58% of its spatial units in these classes. The analysis showcased wide variations in susceptibility across countries, thus highlighting heterogeneity often missed in national datasets. This thereby provides insight into regions and areas within each country with the potential for severe negative outcomes from disease outbreaks and natural or man-made disasters. The datasets presented here are publicly available as part of the

**Data availability statement:** Main data sources (PHIAs) are publicly available but have to be requested to the data curators themselves. The authors of this publication received permission to use the PHIA datasets only for this specific purpose. Therefore, we are unable to share these datasets. However, the PHIA datasets are available to the public upon request, directly to the owner of the data. To access NAIIS 2019, please contact the National Bureau of Statistics [https://microdata.nigerianstat.gov.ng/index.php/catalog/65/related-materials]. To access the BAIS V 2021, please contact Statistics Botswana [https://microdata.statsbots.org.bw:4443/index.php/catalog/26], and for the ZAMPHIA 2021, please contact the Zambian Statistics Agency. The remaining datasets can be found at the following link: https://phia-data.icap.columbia.edu/ The indexes described in these paper are publicly available and can be accessed here: https://zenodo.org/records/15484334.

**Funding:** National Institutes of Health U54TW012041.

**Competing interests:** The authors have declared that no competing interests exist.

INFORM Africa Research project, and provide an evidence base to inform strategic decision-making.

---

## 1. Introduction

In Sub-Saharan Africa, geography shapes not only the physical landscape but also profoundly impacts the resilience fabric of communities. Place susceptibility — a community's capacity to anticipate, respond to, and recover from adverse events — is especially pronounced in regions where factors such as poverty, limited healthcare access, and infrastructure deficiencies converge [1,2]. Place Susceptibility Indexes (PSIs) have proven to be critical tools for identifying areas most in need, enabling governments, health organizations, and development agencies to allocate resources more effectively [3–5]. However, current PSIs for Africa are often calculated at broader scales, such as national or regional levels, which overlook the intricate variations present within smaller communities [6]. By enhancing the spatial resolution of these indexes, governments can gain a clearer, more actionable view of local susceptibilities, making it possible to tailor interventions that align with the specific needs of each community.

A high-resolution PSI would be informative and capable of offering actionable insights at the lower-order administrative level. Such high-granularity data could capture widely varying factors that could impact place susceptibility or resilience. For example, rural communities often suffer from a lack of healthcare facilities, safe drinking water, and education opportunities, which increases their risk when faced with crises like disease outbreaks or extreme weather events. Urban areas, while sometimes better connected, may struggle with issues such as overcrowding, pollution, and inadequate waste management, which also elevate susceptibility. Furthermore, rapid urbanization paired with severe weather has led to changes in the urban thermal environment, making these populations more vulnerable to heat stress [7]. By integrating these diverse factors into PSIs at a finer spatial scale, governments and organizations can better prioritize areas according to local challenges and capacity deficits.

In Sub-Saharan Africa, however, studies on place susceptibility remain scarce. Most PSIs in Africa are calculated at broader scales—national or regional. Thus, overlooking local variations and peculiarities that significantly shape susceptibility. Lawal and Arokoyu [6] provided valuable insights into regional susceptibility from local patterns in Nigeria. However, there remains a data gap for many other countries across Sub-Saharan Africa.

The need for high-resolution PSIs becomes even more critical given the variability of economic and infrastructural conditions across urban and rural areas. For instance, Li, Lewis [8] highlighted how seroprevalence and susceptibility patterns differed across demographic and geographic contexts in the United States, revealing higher infection rates in susceptible and rural areas. This variability is also evident from the findings of Lawal and Osayomi [9], which confirm that vulnerabilities are not

uniformly distributed, making broad-scale assessments insufficient for identifying the nuanced needs of diverse African communities.

This study aims to address this gap by developing high-resolution PSIs for selected African countries, assessing susceptibility at the 3rd-order administrative level (e.g., local government areas (LGA) or districts).

## 2. Materials and methods

### 2.1. Data

**2.1.1. Population-based HIV impact assessment (PHIA).** The Population-based HIV Impact Assessments (PHIA) are nationally representative, cross-sectional, household-based surveys assessing the state of the HIV epidemic in specific affected countries since 2014. The survey is led by each country's Ministry of Health and it is funded by the U.S. President's Emergency Plan for AIDS Relief (PEPFAR) [10]. As part of the interviews, household composition and individual sociodemographic characteristics are collected. Furthermore, as part of this survey, masked geolocation of households are collected [11]. This analysis utilized the most recently completed PHIAs across 10 countries in sub-Saharan Africa: Tanzania (2022), Botswana (2021), Zambia (2021), Eswatini (2021), Mozambique (2021), Malawi (2020), Zimbabwe (2020), Uganda (2020), Lesotho (2020), and Nigeria (2018) [12–21].

**2.1.2. Geocovariates.** The analysis employed eleven covariates for analysis sources and resolution are stated in S1 Table. The Enhanced Vegetation Index (EVI), computed using Google Earth Engine, is a satellite-derived measure of vegetation health. In contrast to the more common Normalized Difference Vegetation Index (NDVI), EVI reduces atmospheric interference and background soil effects, making it particularly suitable in densely vegetated regions [22,23]. The NDVI is also a satellite data-derived measure that quantifies vegetation cover and health according to the differential between near-infrared and red-light reflectance [24]. Both NDVI and EVI are important metrics in monitoring land use change, deforestation, etc.

Land Surface Temperature (LST) records the surface temperature of Earth using satellites. LST illuminates climate variability, urban heat islands [25], and land cover change [26], thus significant in climate adaptation [27] and public health research [28]. In addition to vegetation and temperature, elevation and terrain slope were also incorporated in the study, which were derived from digital elevation models. These factors influence settlement patterns, agricultural suitability, and erosion risk because steep slopes are difficult for infrastructure planning.

Slope (SLOPE) is land surface steepness, derived from digital elevation models (DEM). Slope controls numerous environmental and human processes, including agriculture, construction of infrastructure, and soil erosion control [29]. Steeper slopes are less buildable and more prone to erosion. In addition, elevation (ELEV) is the height above sea level and plays a very significant role in explaining climate variation, flood hazard, and land use planning. Higher altitudes have colder climatic conditions and various plant species than lowland areas [30]. Altitude also influences settlement patterns and agricultural potential [31].

Population data was obtained as Population density in the form of people per pixel (POPP) was obtained from worldpop. org. The variable is used to quantify urbanization trends, resource availability, and access to services. Closely related to it is the accessibility index (ACCESS), which approximates the accessibility of populations to basic services like healthcare, schools, and markets. High values of the accessibility index denote better infrastructure, whereas low values denote shortages of services.

Transportation factors were included to understand mobility and economic activity. Distance to major roads (DMROADS) and distance to major road intersections (DMROADSINT) were considered to understand accessibility and connectivity in the region. Areas closer to roads and intersections have higher chances of more economic opportunities and development [32]. Moreover, distance to water bodies (DMWATER) was included as a significant covariate, considering its significance in water resource management, agriculture, and the estimation of flood risks. Distance to rivers, lakes, or reservoirs tends to influence settlement structure and land use patterns [33].

Finally, nighttime light intensity (NLIGHTS) was included as a surrogate for economic activity and urbanization. From satellite observations, this metric track artificial light emissions during the night, thus serving as an indicative parameter of development and electrification. Higher nighttime lighting intensity is likely to trail urbanization, and areas with low intensity would cover rural or underdeveloped regions [34,35].

## 2.2. Methods

**2.2.1. Data preparation and preprocessing.** The household, roster, and adult surveys from each country's PHIA were examined for place susceptibility-relevant variables. The selected inputs and computed variables for each of the countries are highlighted in S1 Tables.

The selected variables were linked to the cluster centroid, and the proportions and medians of the selected variables were computed for the survey clusters. Household Final Weight, Trimmed Person Nonresponse Adjusted Weight, and Individual Final Weight were used for weighting in the computation of new variables from household, roster, and individual survey data, respectively.

Variance was computed for each of the newly derived variables, and those with zero variance were dropped. The remaining set was subjected to correlation analysis to identify redundant variables using the approach proposed by Lawal [36]. A threshold of 0.7 was set for the elimination of a variable when a pair was highly positively correlated. Thus, one variable was dropped whenever a pair showed a correlation coefficient ≥0.7. Next, factor analysis was carried out to further narrow down the variable set. A sampling adequacy test was carried out to ensure that the remaining dataset was adequate for factor analysis. The principal factor analysis was adopted for factor extraction, while the Varimax rotation was adopted for factor rotation. This approach ensured that the number of high-loading variables for each factor was minimized thus simplifying the interpretation of the factors. The highest loading variables for each factor were selected as representative of each factor. These were subsequently subjected to geostatistical surface creation.

**2.2.2. Geostatistical modeling.** To create the geostatistical surface for each of the high-loading factors selected from section 2.2.1, a Bayesian spatial statistical model was developed using Integrated Nested Laplace Approximation (INLA) in R (Version 4.3). The Bayesian approach was adopted because it can fully account for uncertainty as the entire posterior distribution is computed [37], it is computationally efficient for integrating spatial and non-spatial random effects when modelling with large datasets [38,39] It has the following components:

a. Mesh Construction: A triangulated mesh was constructed for the spatial domain using a country-specific minimum distance between vertices and maximum edge lengths. This mesh formed the basis for spatial random field approximation.

b. Spatial Process Specification: A Matérn spatial process was specified using a Stochastic Partial Differential Equation (SPDE) approach, with α = 2 and a spatial constraint to ensure model identifiability.

c. Model Selection: The study implemented an iterative forward selection procedure to identify the most relevant geocovariates. Starting with an intercept-only model, covariates were sequentially added based on improvements in the Deviance Information Criterion (DIC) and Watanabe-Akaike Information Criterion (WAIC). DIC and WAIC provide relevant information for model selection and comparison balancing model fit and complexity.

d. Response Distribution: The response variable was modeled using a Beta distribution with a logit link function, incorporating censoring values based on the response scale.

Spatial predictions were generated on a regular grid across the study area. The Mean predicted values, Lower (2.5th percentile) prediction intervals, and Upper (97.5th percentile) prediction intervals we generated for each response variable. These predictions were rasterized to create continuous surfaces of the response variable and associated uncertainty

measures. The spatial resolution of the prediction grid was determined by aggregating the original geocovariates raster by a factor of 10 (1 km) to balance computational efficiency and spatial detail. The geocovariates are all at 100m resolution, sources are highlighted in SI-Table 1. Geocovariates were sourced from data repositories of www.worldpop.org, and Google Earth Engine.

The model performance was assessed through multiple diagnostic measures namely:

DIC and WAIC for model comparison

Fixed effects estimates and their credible intervals

Random effects structure

Posterior predictive checks

Cross-validated probability integral transform (PIT) values

**2.2.3. Place susceptibility scores (PSS) and PSI computation.** PSS were computed based on the aggregation of the geostatistical surfaces (each high-loading variable for each country). The maximum and median values for the LGA level, as well as the maximum at the State Level (Boundary just above the LGA), were computed to generate the PSS. These indicators were processed using two distinct approaches based on their directionality.

If a high value confers low susceptibility (HSLS), such as the proportion owning a car/automobile or with better quality roofing material, being employed, and having access to healthcare facilities, the PVS computation involved:

1. Calculating X: the difference between State median and LGA median values

2. Calculating Y: X added to the Maximum of X

3. PSS is thus obtained by normalizing Y, dividing Y by its maximum value

If a high value confers a high-susceptibility (HSHS) indicator, such as the proportion of people reporting sickness and unable to work (in the last three months), the median household size, and the proportion of women currently pregnant, the PSS calculation involved:

1. Calculating X: the ratio of LGA-level and the state-level median values

2. PSS is computed by normalizing X, dividing X by its maximum value

The final PSI was computed as the arithmetic combination of all the PSS.

## 2.3. Validation

To explore the validity of the PSI computed, we adopted the Convergent validity approach. As such, we collated data from the latest version (8.1) of the Subnational Human Development Index archive [40] for each of the countries and the specific year under consideration. The data is available at the State level (equivalent to Level 1 boundary). However, the PSI computed was at Level 2 (2nd Order administrative boundary), thus we computed the median PSI for each of the States (2nd Order administrative boundary) for the validity test. A correlation test was carried out using the Spearman Rank Correlation comparing the health and life expectancy indexes from the Subnational Human Development Index (SDHI) to the state-level aggregated PSI – median. These (health and life expectancy indexes) were selected because of their relevance for health and longevity, which is part of PSI and is also meant to capture. Thus, a negative correlation indicates that the PSI is relatively valid, because high-value HDI indicates good human development, while high PSI indicates higher susceptibility.

## 3. Results

### 3.1. Spatial predictors across selected countries and place susceptibility index components

We developed Bayesian spatial statistical models using Integrated Nested Laplace Approximation (INLA) across the 10 countries for which data was collated. This resulted in the creation of 49 distinct models for various PSI components. Each model incorporated a spatial random effect to account for spatial autocorrelation and unobserved geographical heterogeneity. The iterative search identified the best predictors for each model as presented in Table 1.

Two predictors consistently emerged as important across the countries and components. Among the geocovariates, ACCESS and LST were each found to be significant predictors across 23 models. Others, such as DMROADS, NLIGHTS, DMRAODSINT, and POPPP, were also prominently found across many models. Except for LST, other environment-related predictors such as SLOPE, ELEV, DMWATER, and NVDI were not frequently found to be significant predictors across the 49 models.

### 3.2. PSI distribution

The examination of the distribution of the PSI across 10 countries revealed substantial variations across the PSI values across countries highlights the need for region-specific interventions (S1 Table). Mozambique showed the highest susceptibility, while Uganda exhibited lowest levels of susceptibility. The mean PSI values ranged from a low of 2.528 in Uganda to a high of 4.754 in Mozambique. Thus, highlighting the differences in PSI values across the country. Notably, the median PSI values were generally close to the means, thus suggesting relatively symmetric distributions of PSI within most countries.

In terms of disparities within each country, the standard deviation of PSI varied considerably. Zambia had the highest standard deviation at 0.556, followed by Malawi at 0.394 and Eswatini at 0.362. In contrast, Mozambique exhibited the lowest standard deviation of 0.133, indicating more homogeneous PSI values across the spatial units considered (3rd-order boundary). The skewness of PSI provided insights into the asymmetry of the distributions. Negative skewness, as observed in Zambia (−0.774), Zimbabwe (−0.567), and Lesotho (−0.510), indicated longer left tails, indicating that more of the LGAs had PSI values below the mean. Conversely, positive skewness, as seen in Malawi (1.024), Botswana (0.754), and Uganda (0.636), suggested longer right tails. Thus, indicating that there are more LGAs with PSI values above the mean (Fig 1).

### 3.3. PSI classification distribution across countries

The Jenks Classification was deployed to identify the natural groupings of the PSI values across the LGAs and the pattern formed by classification; this classification is designed to minimize variance within classes and maximize variance between classes, thereby identifying the most natural groupings in the data [41]. The results are presented in Fig 2 while the breakpoints for the classes are presented in S1 Table. For Botswana, the largest proportion of LGAs fell into the "Relatively Low" PSI class (53%), followed by the "Low" class (17%). Each of the "Relatively Moderate" and "Relatively High" classes has 13% of the spatial units, and 3% of the LGAs belong to the "Very High" susceptibility class. This distribution indicates a relatively lower overall susceptibility in Botswana, with a small proportion of areas facing significant challenges.

In the case of Eswatini, a combined 45% of the spatial units considered belong to the "Relatively Low" and "Low" susceptibility classes, suggesting a moderate level of susceptibility across a significant portion of the country. The "Relatively Moderate" class accounted for 33% of the LGAs, while 13% were classified as "Relatively High." The remaining 9% fell into the "Very High" category, highlighting areas with heightened susceptibility.

Lesotho presented a different susceptibility profile, with the "Very High" class comprising the largest proportion of LGAs at 26%. The "Relatively High" class followed closely at 24%, indicating that a significant portion of the country faces elevated susceptibility. The combined "Low" and "Relatively Low" classes accounted for 29% of the LGAs, while the

**Table 1.  PSI Components and Geocovariates for all selected countries.**

| No model | Country | PSI Component | Geocovariates |
|---|---|---|---|
| 1 | Botswana | Better material of roof | NLIGHTS, LST, SLOPE, EVI, ACCESS |
| 2 | | Employed household members | LST, SLOPE, DMROADSINT, POPPP |
| 3 | | Visited health facility for antenatal care | DMROADSINT, POPPP |
| 4 | | Sick for at least 3 months in the past year | LST, NDVI |
| 5 | | Household owns a working car | LST |
| 6 | Eswatini | Better materials for exterior walls | ACCESS, POPPP, DMROADS, DMROADSINT |
| 7 | | Household size | LST, ELEV, POPPP, ACCESS, DMROADSINT, EVI, SLOPE |
| 8 | | Currently married or living with a partner | LST, POPPP, ACCESS |
| 9 | | Visited health facility for antenatal care | LST, ELEV, ACCESS, POPPP, DMROADSINT |
| 10 | | Household owns a working car | ACCESS, ELEV |
| 11 | Lesotho | Household size | ACCESS, LST, ELEV, DMROADS, NDVI |
| 12 | | Sick for at least 3 months in the past year | POPPP, NLIGHTS |
| 13 | | Proportion of women | SLOPE |
| 14 | | Household owns a working car | ACCESS, NLIGHTS, DMROADS |
| 15 | Malawi | Employed household members | LST, NLIGHTS |
| 16 | | Household size | ACCESS, DMROADS, ELEV, LST, EVI |
| 17 | | Household received economic support | SLOPE |
| 18 | | Women currently pregnant | LST, ELEV, DMWATER, NDVI |
| 19 | | Proportion of women | ACCESS, DMROADS, POPPP, EVI |
| 20 | | Household owns a working mobile phone | NLIGHTS, DMROADSINT, EVI, ACCESS |
| 21 | Mozambique | Better toilet facility type | NLIGHTS, LST, DMWATER, DMROADS |
| 22 | | Employed | LST, DMWATER |
| 23 | | Household size | SLOPE |
| 24 | | Visited health facility for antenatal care | LST, NLIGHTS, ACCESS, ELEV |
| 25 | | Proportion of women | ACCESS, DMROADSINT |
| 26 | | Household owns a working car | DMWATER, DMROADSINT, DMROADS, NLIGHTS, LST |
| 27 | Nigeria | Material of household roof | DMROADSINT, ACCESS, LST |
| 28 | | Children per household | LST, ELEV, NLIGHTS, ACCESS |
| 29 | | Proportion of women | LST, DMROADSINT, POPPP, ACCESS |
| 30 | | Household owns a working refrigerator | DMROADSINT, ELEV, LST, DMROADS, POPPP |
| 31 | | Household owns a working motorcycle/scooter | LST, ACCESS, DMROADS |
| 32 | Tanzania | Better floor material | NLIGHTS, DMROADS, DMWATER, DMROADSINT, POPPP |
| 33 | | Employed | DMROADS |
| 34 | | Number of rooms for sleeping | POPPP |
| 35 | | Proportion of women | NLIGHTS, NDVI |
| 36 | | Household owns a working motorcycle/scooter | DMROADSINT |
| 37 | Uganda | Better household water source | NLIGHTS, DMROADS, POPPP, ACCESS, SLOPE, NDVI |
| 38 | | Employed household members | LST, NLIGHTS, DMROADSINT, DMROADS, DMWATER |
| 39 | | Married | ACCESS, ELEV, SLOPE, EVI, LST |
| 40 | | Visited health facility for antenatal care | ELEV, SLOPE, EVI, LST, DMROADS |
| 41 | Zambia | Better floor material | ACCESS |
| 42 | | Employed household members | ACCESS, POPPP, DMWATER, EVI |
| 43 | | Proportion of women | NLIGHTS, DMROADSINT |
| 44 | | Household owns a working mobile phone | ACCESS, SLOPE, NLIGHTS, POPPP |
| 45 | | Household owns a working motorcycle/scooter | NLIGHTS, NDVI |

*(Continued)*

**Table 1.** (Continued)

| No model | Country | PSI Component | Geocovariates |
|---|---|---|---|
| 46 | Zimbabwe | Better toilet facility type | EVI |
| 47 | | Household size | DMROADS, SLOPE |
| 48 | | Married | ACCESS, DMROADS |
| 49 | | Women currently pregnant | SLOPE |

ACCESS (accessibility index), DMROADS (distance to major roads), DMROADSINT (distance to major road intersections), DMWATER (distance to water bodies), ELEV (elevation), *EVI* (*enhanced vegetation index*), *LST* (*land surface temperature*), *NDVI* (*normalized difference vegetation index*), NLIGHTS (nighttime light intensity), POPPP (people per pixel), SLOPE (land surface steepness).

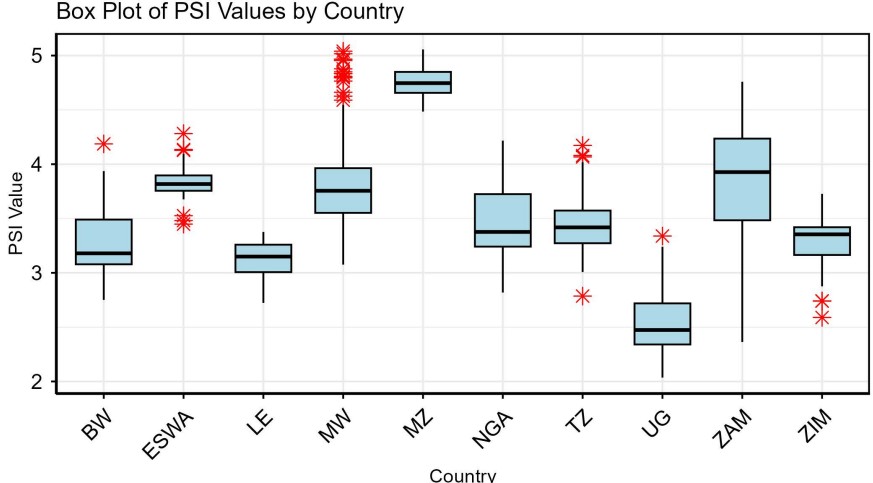

**Fig 1. Box-Whisker Plot of the PSI Values across countries.** *BW* (Botswana), *ESWA*(Eswatini), *LE*(Lesotho), *MW*(Malawi), *MZ*(Mozambique), *NGA*(Nigeria), *TZ*(Tanzania), *UG* (Uganda), *ZAM*(Zambia), *ZIM* (Zimbabwe).

"Relatively Moderate" class made up the remaining 21%. This distribution underscores the prevalence of high susceptibility in Lesotho, with a substantial number of areas requiring targeted interventions to address these challenges.

In Malawi, the majority of LGAs were distributed across the "Relatively Low" and "Relatively Moderate" classes, each accounting for 30% of the spatial units. The "Low" class represented 21% of the LGAs, while the "Relatively High" and "Very High" classes made up 11% and 6%, respectively. This distribution suggests a moderate level of susceptibility across Malawi, with a significant portion of areas experiencing relatively lower susceptibility but still a notable presence of higher susceptibility in certain regions. Four of the spatial units could not be classified due to incomplete data.

The case of Mozambique showed a more balanced distribution across the PSI classes. The "Low" and "Relatively Low" classes each accounted for 21% of the LGAs, while the "Relatively Moderate" and "Relatively High" classes also represented 18% and 21%, respectively. The "Very High" class made up 19% of the LGAs, indicating a substantial presence of areas with significant susceptibility. This distribution highlights the varied susceptibility landscape in Mozambique, with a considerable number of areas facing moderate to high levels of susceptibility.

Tanzania displayed a relatively even spread across the PSI classes. The "Low" and "Relatively Low" classes represented 24% and 25% of the LGAs, respectively, while the "Relatively Moderate" class accounted for 26%. The "Relatively

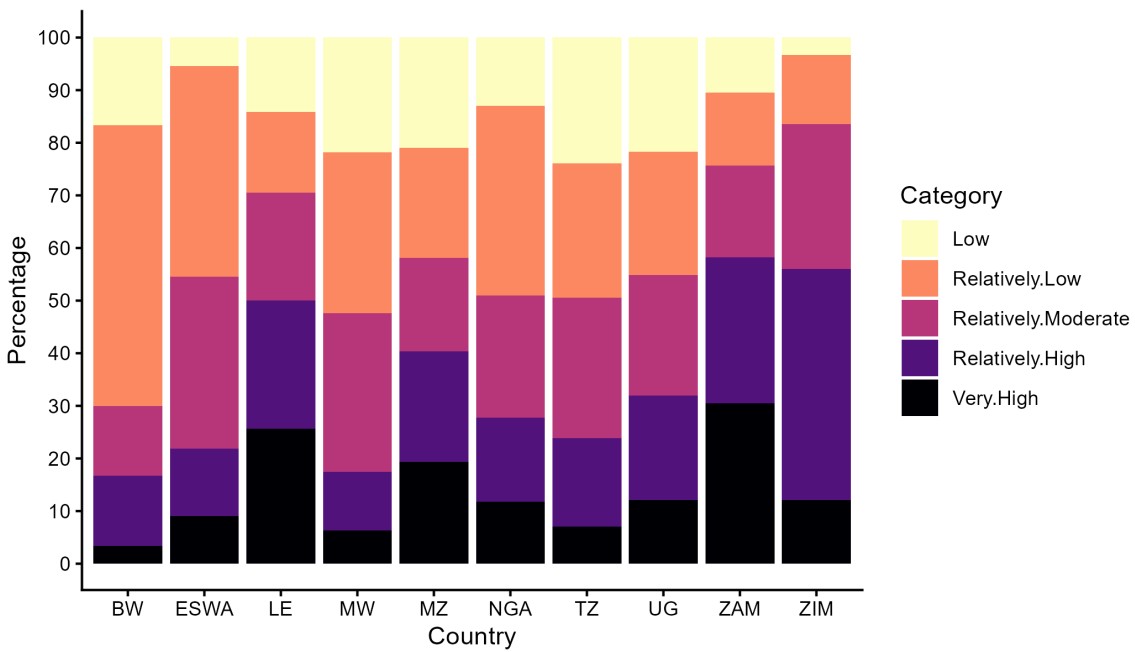

**Fig 2. Distribution of PSI Classes using Jenks Natural Break Classification by country.** *BW* (Botswana), *ESWA*(Eswatini), *LE*(Lesotho), *MW*(Malawi), *MZ*(Mozambique), *NGA*(Nigeria), *TZ*(Tanzania), *UG* (Uganda), *ZAM*(Zambia), *ZIM* (Zimbabwe).

High" and "Very High" classes made up 17% and 7% of the LGAs, with 1% remaining unclassified. This distribution suggests a moderate level of susceptibility across Tanzania, with a significant portion of areas experiencing relatively lower susceptibility, but still some regions facing higher challenges.

The distribution of the classes in Uganda showed that "Low" and "Relatively Low" classes are found across 22% and 23% of the LGAs, respectively. The "Relatively Moderate" and "Relatively High" classes each accounted for 23% and 20% of the LGAs, while 12% fell into the "Very High" category. This distribution indicates a balanced mix of susceptibility levels across Uganda, with a considerable presence of areas in both the lower and higher susceptibility classes.

Zambia showed a higher concentration of LGAs in the "Relatively High" and "Very High" classes, representing 28% and 30% of the spatial units, respectively. The "Relatively Low" and "Low" classes accounted for 14% and 10% of the LGAs, while the "Relatively Moderate" class made up 17%. This distribution highlights the prevalence of high susceptibility in Zambia, with a significant proportion of areas facing elevated challenges.

In Zimbabwe, the highest proportion of LGAs is in the "Relatively High" class at 44%, indicating a substantial presence of areas with elevated susceptibility. The "Relatively Moderate" class accounted for 27% of the LGAs, while the "Low" and "Relatively Low" classes represented 3% and 13%, respectively. The "Very High" class made up 12% of the LGAs. This distribution underscores the significant susceptibility challenges in Zimbabwe, with a large majority of areas falling into the higher susceptibility categories.

For Nigeria, there was a significant proportion of LGAs in the "Relatively Low" class (36%), followed by the "Low" class at 13%. The "Relatively Moderate" and "Relatively High" classes accounted for 23% and 16% of the LGAs, respectively, while 12% fell into the "Very High" category. This distribution indicates a mix of susceptibility levels across Nigeria, with a notable concentration of areas in the lower to moderate susceptibility classes but still a significant presence of highly susceptible regions.

### 3.4. PSI Validation

The results from the correlation analysis are presented in Table 2. The expected direction of association is negative; thus, those with the expected direction of association are grouped as aligned and non-aligned when the association is otherwise.

### 3.4.1. PSI aligned health and life expectancy indexes.

The results for Botswana indicate a strong negative correlation between the median PSI and both the health index and life expectancy, as shown in Table 2. This suggests that higher PSI values are associated with lower health outcomes and reduced life expectancy in this country. The findings align with the expected direction of correlation, thus implying that PSI as computed is capturing most of the relevant factors for social susceptibility leading to negative outcomes for health and life expectancy in the country.

The results for Malawi show weak negative correlations between the median PSI and both health index and life expectancy. This aligns with the expected direction, suggesting that higher PSI values are slightly associated with lower health outcomes and reduced life expectancy. However, the weak strength of the correlations indicates that the PSI may not have captured most of the factors conferring social susceptibility and consequently negative outcomes for health and life expectancy in Malawi.

In Mozambique, the correlations between the median PSI and both the health index and life expectancy are very weak and negative. This suggests that the PSI captured just a fraction of place attributes that could make it susceptible. The direction of association was as expected for the relationship between PSI and health or life expectancy in Mozambique at the State level.

The analysis for Nigeria reveals moderate to strong negative correlations between the median PSI and both the health index and life expectancy. This indicates that higher PSI values are associated with lower health outcomes and reduced life expectancy, consistent with the expected direction of the association. The result suggests that the PSI is valid as computed from the context of life expectancy and health.

For Tanzania, weak negative correlations were found between the median PSI and both the health index and life expectancy. This aligns with the expected direction, suggesting that higher PSI values are slightly associated with lower health outcomes and reduced life expectancy. This confirms that the index is valid, however, it may not have captured many of the other factors that could result in negative outcomes for health and life expectancy at the State level in Tanzania.

### 3.4.2. PSI misaligned with health and life expectancy.

In Eswatini, the analysis reveals strong positive correlations between the median PSI and both the health index and life expectancy. This contrasts with the expected negative

**Table 2. Correlation between 2nd Administrative Level Aggregated PSI values (Median) and Selected Health Development Index (HDI) Component.**

| Country | Correlation Coefficient with HDI Index | |
|---|---|---|
| | **Health Index** | **Life Expectancy Index** |
| Botswana | −0.901 | −0.897 |
| Eswatini | 0.825 | 0.825 |
| Lesotho | 0.349 | 0.638 |
| Malawi | −0.309 | −0.313 |
| Mozambique | −0.058 | −0.059 |
| Nigeria | −0.648 | −0.648 |
| Tanzania | −0.287 | −0.287 |
| Uganda | 0.652 | 0.660 |
| Zambia | 0.150 | 0.150 |
| Zimbabwe | 0.791 | 0.792 |

correlation. This showed the PSI may have captured some of the factors conferring place susceptibility, but the direction of the scoring may not be adequate in ascertaining the expected relationship between health and life expectancy.

For Lesotho, moderate positive correlations were observed between the median PSI and both the health index and life expectancy. This suggests that higher PSI values are associated with improved health and longer life expectancy, contrary to the expected negative relationship. Thus, indicating a similar scoring challenge as noted for Eswatini.

In Uganda, moderate to strong positive correlations were observed between the median PSI and both the health index and life expectancy. This contrasts with the expected negative correlation for a valid PSI. This showed that while factors relevant to place susceptibility have been captured (based on the strength of the correlation), the scoring is not adequate to reflect the expected direction of association for health and life expectancy at the State level in Uganda.

The results for Zambia show very weak positive correlations between the median PSI and both the health index and life expectancy. The direction contrasts with the expected and the strength of the association indicates that many of the factors conferring place susceptibility may not have been captured. Thus, for Zambia at the State level, many of the factors that could confer susceptibility to negative outcomes for health and life expectancy were not captured in the computed PSI.

In Zimbabwe, strong positive correlations were found between the median PSI and both the health index and life expectancy. This indicates that higher PSI values are associated with better health outcomes and longer life expectancy, contrary to the expected negative correlation. This indicates that the computation of the PSI had a similar challenge of scoring as noted for other countries in this category.

The observed positive correlations between PSI and health quality indicators may be attributed to the fact that higher PSI scores are typically found in relatively urbanized or developed areas of these countries, where living conditions may be challenging (e.g., crowding, poverty). These areas also tend to have better access to healthcare facilities, which may relatively improve health outcomes. In addition to this, the aggregation of the PSI may also be responsible for this misalignment. However, since the development of the construct is valid (i.e., factors that could confer susceptibility were correctly identified and utilized), the lack of data to ascertain concurrent validity does not invalidate the PSI computed for these PSI countries. Data for adequate validation is a significant challenge. However, in spite of that, the next best approach is to ascertain construct validity, which was explicitly addressed in this study.

## 4. Discussion

PSI has offered several practical benefits across disaster risk management. The index can estimate the susceptibility of a community to natural and human-made disasters [3,42] or quantitatively describe the relative susceptibility of a place [9,43]. These qualities and their adaptability have enhanced the wide application. Recently, PSI has witnessed a significant expansion of usage [44], and there is an increasing application in public health research and practice to identify and address health disparities.

There is a growing impact of natural disasters because of the increasing severity and frequency of climate-related hazards, leading to escalating disaster losses. The persistence of social and economic inequalities has created disparities in outcomes for public health and hazard events. These, coupled with the need for sustainable development and the increasing relevance of the global health perspective, have made it necessary to understand how place susceptibility can influence outcomes for communities and regions, especially in a changing climate. Thus, detailed and updated information about place susceptibility has supported disaster management and public health efforts in resource allocation, intervention planning, evaluation, identification of susceptible communities, understanding the linkage to chronic diseases and disease outbreaks, and modeling. This study presented the results of an approach for modelling PSI at the 2nd-order administrative boundary using population-based surveys, Bayesian spatial statistical modelling and relevant geocovariates (Fig 3). Thus, providing a tool for decision support in disaster risk management and public health.

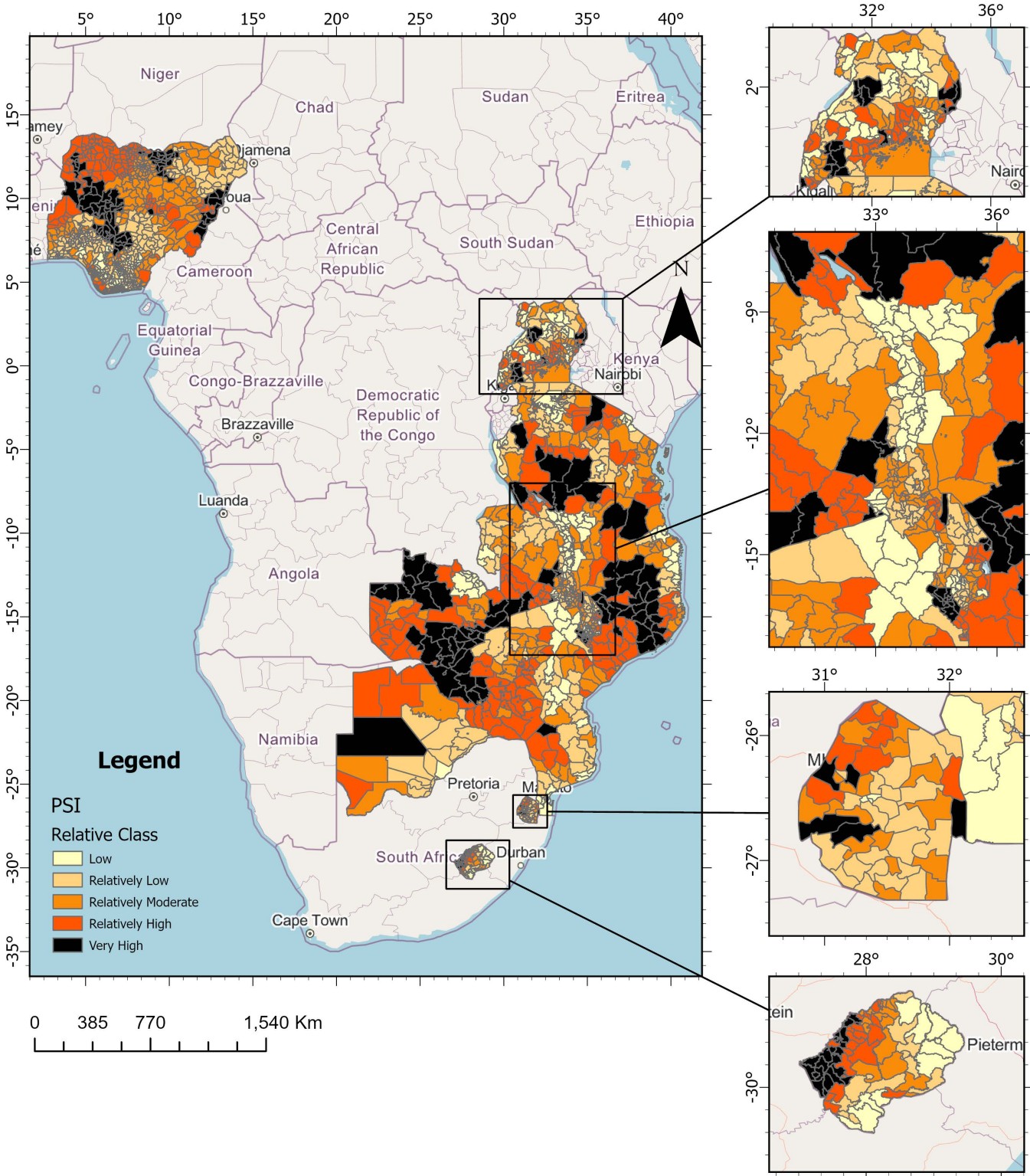

**Fig 3. Distribution of PSI classes using Jenks Natural Break Classification across all countries investigated (Source of Basemap OpenStreetMap).**

To achieve the United Nations Sustainable Development Goals (SDGs) [45], there is a need to understand and quantify place susceptibility across places and communities. Currently, where such quantifications are available in sub-Saharan Africa, they are usually at the National level or the 2nd order administrative units. This is grossly inadequate for location efforts at addressing disaster risk response, preposition, resource allocation for emergencies/disasters, and public health emergencies. With increasing availability and regularity of population surveys and advances in Geocomputation, better and more granular PSI can be developed to support efforts toward achieving the SDGs and other disaster risk management and public health efforts.

PSI is very useful for managing and understanding public health crises, thus helping to understand disparities in outcomes of disease, healthcare, and hazard events. Several studies have highlighted these across different parts of the world. Tortolero et al. found a positive association between PSI and COVID-19 incidence in Haris County, Texas [46]. Blanco et al. confirmed that the COVID-specific PSI is positively correlated with the weighted COVID-19 seroprevalence estimates at LGA-level across four States in Nigeria [47]. This highlighted the usefulness of PSI in prioritizing communities for resource allocation for emergency response and preparedness.

Moore et al. examined associations between county-level stillbirth rates, environmental risk factors for stillbirth, and place susceptibility in the United States between 2015–2018 [48]. They reported a positive association between stillbirth rates and PSI. Another investigation of the relationship between obesity and PSI in the United States highlighted the relevance of PSI in informing about the underlying determinants of obesity [49]. They found that areas with high PSI have a positive relationship with obesity, thus corroborating other studies that have reported that higher PSI is generally related to negative health outcomes. They also reported that high place susceptibility was associated with unhealthy neighborhood environment – lack of access to exercise opportunities, high presence of fast-food restaurants, etc. This thus highlighted the relevance of PSI for the assessment of health issues, e.g., identifying communities with a higher risk of obesity.

In a recent work by Alem et al., they utilized PSI in the development of a mathematical model for optimizing humanitarian supply chains in natural disaster response [50]. The model was tested using a detailed case study based on fourteen years of Brazilian disaster data. The result showed that the benefit of using PSI is more significant as susceptibility increases. Thereby, highlights the value of PSI in preparedness and response to natural disasters. Others also have examined the relationship between place susceptibility and flood-related death and property damage in the USA [51]. They revealed that explained variance increases with models adding PSI, which is significantly correlated with damage ratios. In addition, they found that PSI explains some variance in outcomes better than biophysical factors alone, and there is strong support that PSI is correlated with higher death and damage in non-coastal flood events in the USA. These provide further credence to the ability of PSI to capture underlying place factors leading to varying resilience and susceptibility in the case of natural hazards events and the potential of PSI to support decision-making in response to such situations.

The dataset developed in this study provides an objective approach for quantifying place susceptibility. Thus, provides a measure for monitoring changes in place susceptibility, evaluating the progress of interventions to address it and reducing the negative impact on health outcomes due to disasters and public health hazards. Place susceptibility represents an important factor that could moderate the impact of the risk communities are exposed to. Therefore, understanding and quantifying it is vital to decipher the capacity to respond and recover from various hazard events [9,52].

In addition to the index, PSS for each component of the index were provided. These are as important as the index in helping to understand the variabilities observed. The work of Acharya and Porwal [53] also reiterated this, and they found that there is a specific PSI component alignment with COVID-19 cases in India. They concluded that it is important to consider PSI domains and not only the overall PSI when managing public health crises.

Our work is limited by the different sources of errors and uncertainties associated with the PSI Index described here. Uncertainties could be introduced from the (i) survey data (ii) geocovariates and, (iii) modelling operations. The credible intervals for the modelled PSI inputs (i.e., variables for the PHIA dataset) were computed to summarize these

uncertainties. Thus, credible ranges of the PSS and the overall PSI were also computable. Future work will focus on expanding spatial (increase countries covered) and temporal (time covered) coverage to create a longitudinal dataset for trend analysis. There is also the need to explore other weighting mechanisms for the computation of the PSI apart from the equal weighting used in this work.

Knowledge of Place Susceptibility is important for managing disasters and public health. Access to granular, actionable data and the capacity to develop and utilize such data remain barriers to data-driven policymaking across many parts of sub-Saharan Africa. Here we have developed a 3rd order geography PSI dataset for countries across sub-Saharan Africa that represents a first-of-its-kind in the region. Thus, combining population-based survey data, geocovariates, and spatial modelling to generate a locally relevant PSI. Our findings showcased wide variations in susceptibility across countries, thus highlighting heterogeneity often missed in national datasets. This thereby provides insight into regions and areas within each country with the potential for severe negative outcomes from disease outbreaks and natural or man-made disasters. The PSI datasets are freely available as a product of the Inform Africa project and can be downloaded here: https://zenodo.org/records/15484334. Governments could use these datasets to inform decision and policy making prioritizing health interventions or emergency response by easily categorizing areas based on local challenges and capacity deficits.

## Supporting information

**S1 Tables. Geospatial Covariates; Computed Variables and Indicators for each Country; Descriptive Statistics for the PSI; and Breakpoint and Bounds of the PSI Classes.**
(DOCX)

## Acknowledgments

We acknowledge the efforts of the PHIA project in collecting and sharing these rich datasets. We also acknowledge the technical support of Danielle Sharp, Rickie Malaba, and Tendayi Mharadze from the US Centers for Disease Control and Prevention Global Health Center.

**Authorship for the INFORM Africa Research Study Group for National Institutes of Health (NIH) D-SI Africa Consortium**

*Akros: Christina Riley, Anna Winters. Centre for the AIDS Programme of Research in South Africa (CAPRISA): Vivek Naranbhai, Felix Made, Salim Abdool Karim. Consortium for Advanced Research Training in Africa (CARTA): Kennedy Otwombe. Institute of Human Virology Nigeria: Alash'le Abimiku, Sophia Osawe, James Onyemata, Patrick Dakum, Fati Murtala-Ibrahim, Nifarta Andrew, Aminu Musa, Tolulope Adenekan, Kenneth Ewerem, Victoria Etuk. Stellenbosch University/ Centre for Epidemic Response and Innovation (CERI):Tulio de Oliveira, Cheryl Baxter, Eduan Wilkinson, Houriiyah Tegally, Jenicca Poongavanan, Michelle Parker, Danilo Silva, Joicymara S. Xavier. University of Maryland Baltimore: Kristen A. Stafford, Manhattan Charurat, Natalia Blanco, Timothy O'Connor, Meagan Fitzpatrick, Mohammad M. Sajadi. University of Port Harcourt. Olanrewaju Lawal. Mmedorenyin Okon Villanova University: Chenfeng Xiong, Weiyu Luo, Xin Wu.*

## Author contributions

**Conceptualization:** Olanrewaju Lawal, Kristen A. Stafford.

**Data curation:** Olanrewaju Lawal, Mmedorenyin Okon, Kristen A. Stafford.

**Formal analysis:** Olanrewaju Lawal, Mmedorenyin Okon.

**Funding acquisition:** Olanrewaju Lawal, Alash'le Abimiku, Kristen A. Stafford.

**Methodology:** Olanrewaju Lawal.

**Writing – original draft:** Olanrewaju Lawal, Mmedorenyin Okon, Natalia Blanco.

**Writing – review & editing:** Natalia Blanco, Christina Riley, James Onyemata, Anna Winters, Timothy O'Connor, Chenfeng Xiong, Alash'le Abimiku, Manhattan Charurat, Kristen A. Stafford.

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
