## [Decision Letter · Decision Letter 0]

2 Oct 2025

Dear Dr. Lawal,

Thank you for submitting your manuscript to PLOS ONE. After careful consideration, we feel that it has merit but does not fully meet PLOS ONE’s publication criteria as it currently stands. Therefore, we invite you to submit a revised version of the manuscript that addresses the points raised during the review process.

Dear authors,

The reviewers have carefully evaluated your manuscript, and both recommend that it be revised before further consideration. I invite you to submit a revised version that addresses all reviewer comments in detail. Please also include a point-by-point response letter outlining how you have handled each comment.

We look forward to receiving your revised submission.

Best regards,

We look forward to receiving your revised manuscript.

Kind regards,

Armin Moghimi

Academic Editor

PLOS ONE

Journal Requirements:

2. Please update your submission to use the PLOS LaTeX template. The template and more information on our requirements for LaTeX submissions can be found at http://journals.plos.org/plosone/s/latex .

3. Please note that PLOS One has specific guidelines on code sharing for submissions in which author-generated code underpins the findings in the manuscript. In these cases, we expect all author-generated code to be made available without restrictions upon publication of the work. Please review our guidelines at https://journals.plos.org/plosone/s/materials-and-software-sharing#loc-sharing-code and ensure that your code is shared in a way that follows best practice and facilitates reproducibility and reuse.

[National Institutes of Health

U54TW012041].

5. Thank you for stating the following in your manuscript:

[This research was funded by the National Institutes of Health, grant number U54TW012041.]

[National Institutes of Health

U54TW012041]

4. In the online submission form, you indicated that [Main data sources (PHIAs) are publicly available but have to be requested to the data curators themselves. The indexes described in these paper are publicly available and can be accessed here: https://zenodo.org/records/15484334].

6. One of the noted authors is a group or consortium [The INFORM Africa Study Group]. In addition to naming the author group, please list the individual authors and affiliations within this group in the acknowledgments section of your manuscript. Please also indicate clearly a lead author for this group along with a contact email address.

7. Please upload a new copy of Figures 1 and 2 as the details are not clear. Please follow the link for more information: https://blogs.plos.org/plos/2019/06/looking-good-tips-for-creating-your-plos-figures-graphics/

8. We note that Figure 3 in your submission contains map images which may be copyrighted. All PLOS content is published under the Creative Commons Attribution License (CC BY 4.0), which means that the manuscript, images, and Supporting Information files will be freely available online, and any third party is permitted to access, download, copy, distribute, and use these materials in any way, even commercially, with proper attribution. For these reasons, we cannot publish previously copyrighted maps or satellite images created using proprietary data, such as Google software (Google Maps, Street View, and Earth). For more information, see our copyright guidelines: http://journals.plos.org/plosone/s/licenses-and-copyright.

1. You may seek permission from the original copyright holder of Figure 3 to publish the content specifically under the CC BY 4.0 license.

Additional Editor Comments:

Dear authors,

The reviewers have carefully evaluated your manuscript, and both recommend that it be revised before further consideration. I invite you to submit a revised version that addresses all reviewer comments in detail. Please also include a point-by-point response letter outlining how you have handled each comment.

We look forward to receiving your revised submission.

Best regards,

Reviewers' comments:

Reviewer's Responses to Questions

**Comments to the Author**

1. Is the manuscript technically sound, and do the data support the conclusions?

Reviewer #1: Yes

Reviewer #2: Yes

2. Has the statistical analysis been performed appropriately and rigorously?

Reviewer #1: Yes

Reviewer #2: N/A

3. Have the authors made all data underlying the findings in their manuscript fully available?

Reviewer #1: Yes

Reviewer #2: Yes

4. Is the manuscript presented in an intelligible fashion and written in standard English?

Reviewer #1: Yes

Reviewer #2: No

Reviewer #1: The manuscript offers a valuable approach to improving the Place Susceptibility Index (PSI) at the local government level in Sub-Saharan Africa. The study is well-organized and addresses a key issue: providing more detailed data on susceptibility, which can help in better managing public health and disaster response efforts. The methodology, which uses Bayesian spatial statistical modeling combined with PHIA data and geospatial factors, is strong and supports the findings effectively.

The results show clear differences in susceptibility across the 10 countries, helping to identify areas that are most vulnerable to disease outbreaks and natural disasters. The classification method used is helpful in uncovering important local variations that national datasets may overlook.

However, there are some areas that could be improved. The abstract, while thorough, could be shortened slightly for better clarity. Some of the references, particularly related to statistical methods and geospatial analysis, might need to be updated to reflect the latest research.

The discussion ties the results to practical applications, but it would be useful to offer more specific recommendations on how policymakers can use PSI. A more detailed conclusion, highlighting the broader impact of this work on Sustainable Development Goals (SDGs), would strengthen the manuscript.

Overall, this is a strong study that could make a significant impact with some minor revisions. It provides a useful tool for public health and disaster management in Africa.

Reviewer #2: The study is interesting, but it lacks proper validation, and the reliability of the findings in relation to the data screening process is questionable.

Comments

The manuscript is interesting and focuses on developing high-resolution PSIs for selected African countries. However, some revisions are necessary to strengthen the work. I recommend the authors consider the following major revisions and clarifications:

-Include more appropriate and precise keywords.

- Incorporate recent studies on PSI, especially those linking urban resilience with rapid urbanization, and clearly highlight the research gap connected to the novelty of this work. The introduction should explicitly present the key research questions, critical knowledge gaps, and how this study addresses them, supported with relevant references.

- The paper does not clearly explain how sampling strategies were employed to address variability. Provide more detail on data preprocessing steps (e.g., handling outliers, normalization, and sampling strategy).

- Expand on the processing details, including data sources, category types, spatial resolution, cloud filtering methods, calculations performed, and software used.

- Discuss the advantages of applying the Bayesian spatial statistical model prior to its use in this study. The authors could consider applying feature sensitivity or feature selection techniques to enhance the model’s application and improve the reliability of the findings.

- Justify the rationale for selecting the Jenks Classification method.

- Address the uncertainty of PSI model outputs. To improve robustness, incorporate uncertainty analyses such as standard deviation, coefficient of variation, Monte Carlo simulations, or entropy measures for each crop type, enabling spatial quantification of prediction uncertainty.

- Provide policy recommendations for areas classified as high and low Place Susceptibility zones, linking them to strategies for improving urban sustainability indicators. Provide an analysis of the relationship between high purchasing power parity and the life expectancy and birth rate across the selected provinces, and explain how these relate to the PSI outcomes.

- Add a dedicated subsection. While some limitations are noted, it would be valuable to include a summary table highlighting current gaps and possible future solutions.

- Revise the conclusion to reconstruct the key findings and emphasize how the outputs can be practically applied by government agencies, policymakers, and urban planners.

**Do you want your identity to be public for this peer review?** For information about this choice, including consent withdrawal, please see our Privacy Policy

Reviewer #1: **Yes:** Clara Ekpekose Oguji

Reviewer #2: No

---

## [Author Response · Author response to Decision Letter 1]

7 Nov 2025

Our responses are documented point-by-point below, and corresponding changes have been marked using tracked changes in the revised version of the manuscript. We have also included a clean copy of the revised manuscript.

Please do not hesitate to contact us if you have any additional questions.

Reviewer #1

1. Clarification of Misalignments in Correlations. Section: Results, PSI Misaligned with Health and Life Expectancy. Issue: The manuscript mentions that countries like Eswatini, Lesotho, and Uganda show positive correlations with health and life expectancy, which contrasts with the expected negative correlation. Revision: Add a sentence or paragraph explaining why these positive correlations might have occurred. Could there be unique contextual factors in these countries that skew the expected outcome? For example, do these countries have better-than-expected healthcare access or infrastructure that could influence the relationship?

Thank you for your comment. We agree with the reviewer. We have provided further clarification in section 3.4.2. We included a paragraph explaining potential reasons for the observed relationship

2. Simplification of Statistical Terminology: Section: Methods, Geostatistical Modeling. Issue: Terms like Deviance Information Criterion (DIC) and Watanabe-Akaike Information Criterion (WAIC) might be unclear to a broader audience. Revision: Simplify or briefly explain the terms in the text or add a footnote.

Thank you for your comment. We have added a brief explanation of the purpose of using these criteria. In section 2.2.2 , we added the following sentence: “DIC and WAIC provide relevant information for model selection and comparison, balancing model fit and complexity.”

3. Improvement of Figure and Table Labels. Section: Results, PSI Distribution and PSI Classification Distribution Across Countries. Issue: The figure captions could be more descriptive to help readers interpret the data. Revision: Add annotations or clarifying notes to the figures where necessary. For example, in Figure 2, explain briefly why certain countries have higher proportions in the "Very High" PSI class.

We agree with the reviewer that the labels for figures and tables help readers understand the figures/tables on their own. We have slightly edited our figures and tables labels. However, it should be noted that because we adopted a particular classification scheme it would with the problem modifiable area unit problem (MAUP) it is not necessary to draw conclusion here it is better to report the distribution. Thus, it would be highly inappropriate to suggest reasons why, but the ultimate is that the results reflected the underlying vulnerability or susceptibility of the places (boundaries considered). It is expected that there would be variation and why some have more than others is directly related to the conditions within the country

4. Streamline the Abstract. Section: Abstract. Issue: The abstract is slightly long and contains some redundancy. Revision: Trim some details while keeping key findings intact. "The Place Susceptibility Index (PSI) has the potential to be a critical tool for Ministries of Health and Public Health and disaster management organizations to prepare for and manage infectious disease outbreaks or natural disasters in Africa." Revised: "The Place Susceptibility Index (PSI) is a critical tool for preparing for infectious disease outbreaks and disasters in Africa.". Simplify technical descriptions in the abstract, e.g., "Jenk’s natural grouping algorithm" could be simplified to "a classification method."

We have edited the abstract to reduce word count and avoid redundancy as suggested.

5. Section: Results, PSI Distribution. Issue: The results section dives into statistical details immediately after the introductory sentence. Revision: Start with a summary of key findings in simpler terms before diving into statistical specifics. Example: "The PSI varied significantly across the 10 countries, with countries like Botswana showing lower susceptibility, while Zambia exhibited higher levels of susceptibility. This variation highlights the need for region-specific interventions ." Then, proceed with the statistical details (e.g., means, medians).

We appreciate the feedback. We have edited section 3.2 of the results section.

6. Minor Typographical Corrections Section: Throughout the manuscript Issue: Some inconsistency in the use of abbreviations and formatting (e.g., PSI vs. PSS, LGA vs. local government areas). Revision: Ensure consistency with the use of abbreviations. For example, PSI should always be used the same way, and LGA should be expanded at first use and followed consistently (e.g., "Local Government Area (LGA)").

We appreciate the reviewer’s feedback. We have proofread the manuscript, ensuring consistency across acronyms. However, PSI and PSS are distinct terms, as described in section 2.2.3. The distinction between the terms is important when describing the methods.

7. Enhanced Discussion on Policy Implications Section: Discussion, PSI and Public Health Interventions. Issue: The manuscript discusses PSI’s usefulness but lacks specific policy recommendations. Revision: Expand the discussion by providing clear recommendations for action. Example: "Governments could use PSI to prioritize health interventions, focusing on regions with high susceptibility to outbreaks. This could include targeted vaccination campaigns or improvements in healthcare infrastructure."

We have expanded on the policy applications of the PSI in our discussion section.

8. Reference: It’s important to make sure that the references in the manuscript, especially those about statistical methods and geospatial analysis, are current and still relevant. For example, the manuscript mentions using Bayesian modeling and other statistical methods. It’s a good idea to check that the studies cited for these methods are up-to-date, as newer research may offer better techniques or improved understanding of these methods. The manuscript also mentions using geospatial tools like Enhanced Vegetation Index (EVI) and Land Surface Temperature (LST). These tools are constantly improving, so it’s important to verify that the sources cited are the most recent ones, as new technologies or research might offer better ways to analyze this data. The study uses Population-based HIV Impact Assessments (PHIA) data, which is often updated. If the manuscript cites older versions of this data, it would be helpful to update the reference to the most recent version, so the analysis is based on the latest available information. Additionally, it may be useful to check if any recent review articles or meta-analyses are available. These sources can summarize the latest research and give a broader view of the topic, which might strengthen the manuscript’s arguments. Updating the references ensures the manuscript is based on the latest research and is more credible.

Thank you for the feedback. The most recent available data, including geocovariates and PHIA surveys, were utilized for the estimation of the PSIs. The PHIA survey data utilized were the most recent available (Section 2.1.1), details of the geocovariates were further added to the Supplementary document. Our work filled a data gap, as such meta-analyses and reviews are not particularly relevant in this case. We have searched for new related literature, but no further pertinent references were identified.

Reviewer #2

1. The manuscript is interesting and focuses on developing high-resolution PSIs for selected African countries. However, some revisions are necessary to strengthen the work. I recommend the authors consider the following major revisions and clarifications. Include more appropriate and precise keywords.

Thank you for the positive feedback. We have added more appropriate and precise keywords.

2. Incorporate recent studies on PSI, especially those linking urban resilience with rapid urbanization, and clearly highlight the research gap connected to the novelty of this work. The introduction should explicitly present the key research questions, critical knowledge gaps, and how this study addresses them, supported with relevant references.

We have added further references, particularly for the context of interest and to highlight the gap this work aims to fill. This work identified a data gap that it is addressing, we justified that in the introduction.

3. The paper does not clearly explain how sampling strategies were employed to address variability. Provide more detail on data preprocessing steps (e.g., handling outliers, normalization, and sampling strategy).

Section 2.2.1 and 2.2.3 already includes the requested details. However, we have further added references to guide the reader to further information.

4. Expand on the processing details, including data sources, category types, spatial resolution, cloud filtering methods, calculations performed, and software used.

The methods section already includes the requested details. However, we have further references to guide the reader to further information. Additionally, we have added information on the data sources and resolution in Section 2.2.2. and table 1 of the supplementary document.

5.Discuss the advantages of applying the Bayesian spatial statistical model prior to its use in this study. The authors could consider applying feature sensitivity or feature selection techniques to enhance the model’s application and improve the reliability of the findings.

We have added justification for Bayesian approach to section 2.2.2, and the adopted feature selection method was referenced with appropriate citation in section 2.2.1

6. Justify the rationale for selecting the Jenks Classification method.

We have added further details on the rationale for selecting the Jenks Classification method in section 3.3.

7.Address the uncertainty of PSI model outputs. To improve robustness, incorporate uncertainty analyses such as standard deviation, coefficient of variation, Monte Carlo simulations, or entropy measures for each crop type, enabling spatial quantification of prediction uncertainty.

The INLA model used for this modelling generated posterior distribution for each of the components of the PSI, the Markov chain Monte Carlo (MCMC) method was used to compute posterior marginal distributions. Table 13 of the supplementary document has the SD and other descriptive statistics.

8.Provide policy recommendations for areas classified as high and low Place Susceptibility zones, linking them to strategies for improving urban sustainability indicators. Provide an analysis of the relationship between high purchasing power parity and the life expectancy and birth rate across the selected provinces and explain how these relate to the PSI outcomes.

We appreciate the reviewer’s feedback. Although we have expanded on the policy implications in our discussion section. However, we consider the relationship between high purchasing power parity and the life expectancy and birth rate across the selected provinces/states, and how this relates to the PSI outcomes is outside the scope of this paper.

9.Add a dedicated subsection. While some limitations are noted, it would be valuable to include a summary table highlighting current gaps and possible future solutions.

We have added further future solutions to the section

10. Revise the conclusion to reconstruct the key findings and emphasize how the outputs can be practically applied by government agencies, policymakers, and urban planners.

---

## [Decision Letter · Decision Letter 1]

28 Dec 2025

Place Susceptibility Index Mapping at Local Government Scale from Population-based Survey for Sub-Saharan Africa

PONE-D-25-28111R1

Dear Dr. Lawal,

We’re pleased to inform you that your manuscript has been judged scientifically suitable for publication and will be formally accepted for publication once it meets all outstanding technical requirements.

Kind regards,

Armin Moghimi

Academic Editor

PLOS One

Additional Editor Comments (optional):

Dear Authors,

I am pleased to inform you that we have received positive feedback from the reviewers on your revisions.

Reviewers' comments:

Reviewer's Responses to Questions

**Comments to the Author**

Reviewer #1: All comments have been addressed

Reviewer #2: All comments have been addressed

2. Is the manuscript technically sound, and do the data support the conclusions?

Reviewer #1: Yes

Reviewer #2: Yes

3. Has the statistical analysis been performed appropriately and rigorously?

Reviewer #1: Yes

Reviewer #2: Yes

4. Have the authors made all data underlying the findings in their manuscript fully available?

Reviewer #1: Yes

Reviewer #2: Yes

5. Is the manuscript presented in an intelligible fashion and written in standard English?

Reviewer #1: Yes

Reviewer #2: Yes

Reviewer #1: The authors addressed all the comments and provided a good description of how and where they inserted the new information.

Reviewer #2: The authors are correctly answered the reviewers' comments and suggestions, and now the article has been accepted

**Do you want your identity to be public for this peer review?** For information about this choice, including consent withdrawal, please see our Privacy Policy

Reviewer #1: **Yes:** Clara Ekpekose Oguji

Reviewer #2: **Yes:** Chiranjit Singha

---

## [Editor Report · Acceptance letter]

PONE-D-25-28111R1

PLOS One

Dear Dr. Lawal,

I'm pleased to inform you that your manuscript has been deemed suitable for publication in PLOS One. Congratulations! Your manuscript is now being handed over to our production team.

Kind regards,

on behalf of

Dr. Armin Moghimi

Academic Editor

PLOS On